# Constrained-CNN losses for
# weakly supervised segmentation

**Hoel Kervadec**
ÉTS Montréal
`hoel.kervadec.1@etsmtl.net`

**Jose Dolz**
ÉTS Montréal

**Meng Tang**
University of Waterloo
Department of computer science

**Éric Granger**
ÉTS Montréal

**Yuri Boykov**
University of Waterloo
Department of computer science

**Ismail Ben Ayed**
ÉTS Montréal

## Abstract

Weak supervision, e.g., in the form of partial labels or image tags, is currently attracting significant attention in CNN segmentation as it can mitigate the lack of full and laborious pixel/voxel annotations. Enforcing high-order (global) inequality constraints on the network output, for instance, on the size of the target region, can leverage unlabeled data, guiding training with domain-specific knowledge. Inequality constraints are very flexible because they do not assume exact prior knowledge. However, constrained Lagrangian dual optimization has been largely avoided in deep networks, mainly for computational tractability reasons. To the best of our knowledge, the method of Pathak et al. [17] is the only prior work that addresses deep CNNs with linear constraints in weakly supervised segmentation. It uses the constraints to synthesize fully-labeled training masks (proposals) from weak labels, mimicking full supervision and facilitating dual optimization.

We propose to introduce a differentiable term, which enforces inequality constraints directly in the loss function, avoiding expensive Lagrangian dual iterates and proposal generation. From constrained-optimization perspective, our simple approach is not optimal as there is no guarantee that the constraints are satisfied. However, surprisingly, it yields *substantially* better results than the Lagrangian-based constrained CNNs in [17], while reducing the computational demand for training. In the context of cardiac images, we reached a segmentation performance close to full supervision using a fraction of the full ground-truth labels (0.1%). While our experiments focused on basic linear constraints such as the target-region size and image tags, our framework can be easily extended to other non-linear constraints, e.g., invariant shape moments [5] or other region statistics [8]. Therefore, it has the potential to close the gap between weakly and fully supervised learning in semantic medical image segmentation. Our code is publicly available.

## 1   Introduction

In the recent years, deep convolutional neural networks (CNNs) have been dominating semantic segmentation problems, both in computer vision and medical imaging, achieving ground-breaking performances when full-supervision is available [10, 11]. In semantic segmentation, full supervision requires laborious pixel/voxel annotations, which may not be available in a breadth of applications, more so when dealing with volumetric data. Therefore, weak supervision with partial labels, for instance, bounding boxes [19], points [1], scribbles [22, 21, 9], or image tags [17, 14], is attracting significant research attention. Imposing prior knowledge on the the network's output in the form

1st Conference on Medical Imaging with Deep Learning (MIDL 2018), Amsterdam, The Netherlands.

of unsupervised loss terms is a well-established approach in machine learning [23, 3]. Such priors can be viewed as regularization terms that leverage unlabeled data, embedding domain-specific knowledge. For instance, the recent studies in [22, 21] showed that direct regularization losses, e.g., dense conditional random field (CRF) or pairwise clustering, can yield outstanding results in weakly supervised segmentation, reaching almost full-supervision performances in natural image segmentation; see the results in [22]. Surprisingly, such a principled direct-loss approach is not common in weakly supervised segmentation. In fact, most of the existing techniques synthesize fully-labeled training masks (proposals) from the available partial labels, mimicking full supervision [19, 14, 9, 6]. Typically, such proposal-based techniques iterate two steps: CNN learning and proposal generation facilitated by dense CRFs and fast mean-field inference [7], which are now the de-facto choice for pairwise regularization in semantic segmentation algorithms.

Our purpose here is to embed high-order (global) inequality constraints on the network output directly in the loss function, so as to guide learning. For instance, assume that we have some prior knowledge on the size (or volume) of the target region, e.g., in the form of lower and upper bounds on size, a common scenario in medical image segmentation [13, 4]. Let $I : \Omega \subset \mathbb{R}^{2,3} \to \mathbb{R}$ denotes a given training image, with $\Omega$ a discrete image domain and $|\Omega|$ the number of pixels/voxels in the image. $\Omega_L \subseteq \Omega$ is a weak (partial) ground-truth segmentation of the image, taking the form of a partial annotation of the target region, e.g., a few points (see Fig. 2). In this case, one can optimize a *partial* cross-entropy loss subject to inequality constraints on the network outputs [17]:

$$\min_{\boldsymbol{\theta}} \ \mathcal{H}(S) \quad \text{s.t} \quad a \leq \sum_{p \in \Omega} S_p \leq b \tag{1}$$

where $S = (S_1, \ldots, S_{|\Omega|}) \in [0, 1]^{|\Omega|}$ is a vector of softmax probabilities[1] generated by the network at each pixel $p$ and $\mathcal{H}(S) = -\sum_{p \in \Omega_L} \log(S_p)$ Priors $a$ and $b$ denote the given upper and lower bounds on the size (or cardinality) of the target region. Inequality constraints of the form in (1) are very flexible because they do not assume exact knowledge of the target size, unlike [25, 2]. Also, multiple instance learning (MIL) constraints [17], which enforce image-tag priors, can be handled by constrained model (1). Image tags are a form of weak supervision, which enforce the constraints that a target region is present or absent in a given training image [17]. They can be viewed as particular cases of the inequality constraints in (1). For instance, a suppression constraint, which takes the form $\sum_{p \in \Omega} S_p \leq 0$, enforces that the target region is not in the image. $\sum_{p \in \Omega} S_p > 1$ enforces the presence of the region.

Even though constraints of the form (1) are linear (and hence convex) with respect to the network outputs, constrained problem (1) is very challenging due to the non-convexity of CNNs. One possibility would be to minimize the corresponding Lagrangian dual. However, as pointed out in [17, 12], this is computationally intractable for semantic segmentation networks involving millions of parameters; one has to optimize a CNN within each dual iteration. In fact, constrained optimization has been largely avoided in deep networks [20], even thought some Lagrangian techniques were applied to neural networks a long time before the deep learning era [24, 18]. These constrained optimization techniques are not applicable to deep CNNs as they solve large linear systems of equations. The numerical solvers underlying these constrained techniques would have to deal with matrices of very large dimensions in the case of deep networks [12].

To the best of our knowledge, the method of Pathak et al. [17] is the only prior work that addresses constrained deep CNNs in weakly supervised segmentation. It uses the constraints to synthesize fully-labeled training masks (proposals) from the available partial labels, mimicking full supervision, which avoids intractable dual optimization of the constraints when minimizing the loss function. The main idea of [17] is to model the proposals via a latent distribution. Then, they minimize a KL divergence, encouraging the softmax output of the CNN to match the latent distribution as closely as possible. Therefore, they impose constraints on the latent distribution rather than on the network output, which facilitates Lagrangian dual optimization. This decouples stochastic gradient descent learning of the network parameters and constrained optimization: The authors of [17] alternate between optimizing w.r.t the latent distribution, which corresponds to proposal generation subject to the constraints[2], and standard stochastic gradient descent for optimizing w.r.t the network parameters.

---

[1]The softmax probabilities take the form: $S_p(\boldsymbol{\theta}, I) \propto \exp f_p(\boldsymbol{\theta}, I)$, where $f_p(\boldsymbol{\theta}, I)$ is a real scalar function representing the output of the network for pixel $p$. For notation simplicity, we omit the dependence of $S_p$ on $\boldsymbol{\theta}$ and $I$ as this does not result in any ambiguity in the presentation.

[2]This sub-problem is convex when the constraints are convex.

We propose to introduce a differentiable term, which enforces inequality constraints (1) directly in the loss function, avoiding expensive Lagrangian dual iterates and proposal generation. From constrained optimization perspective, our simple approach is not optimal as there is no guarantee that the constraints are satisfied. However, surprisingly, it yields *substantially* better results than the Lagrangian-based constrained CNNs in [17], while reducing the computational demand for training. In the context of cardiac image segmentation, we reached a performance close to full supervision while using a fraction of the full ground-truth labels (0.1%). Our framework can be easily extended to non-linear inequality constraints, e.g., invariant shape moments [5] or other region statistics [8]. Therefore, it has the potential to close the gap between weakly and fully supervised learning in semantic medical image segmentation. Our code is publicly available [3].

## 2    Proposed loss function

We propose the following loss for weakly supervised segmentation:

$$\mathcal{H}(S) + \lambda \mathcal{C}(V_S), \tag{2}$$

where $V_S = \sum_{p \in \Omega} S_p$, $\lambda$ is a positive constant that weighs the importance of the constraints and function $\mathcal{C}$ is given by (See the illustration in Fig. 1):

$$\mathcal{C}(V_S) = \begin{cases} (V_S - a)^2, & \text{if } V_S \leq a \\ (V_S - b)^2, & \text{if } V_S \geq b \\ 0, & \text{otherwise} \end{cases} \tag{3}$$

Now, our differentiable term $\mathcal{C}$ accommodates standard stochastic gradient descent. During back-propagation, the term of gradient-descent update corresponding to $\mathcal{C}$ can be written as follows:

$$-\frac{\partial \mathcal{C}(V_S)}{\partial \boldsymbol{\theta}} \propto \begin{cases} (a - V_S) \frac{\partial S_p}{\partial \boldsymbol{\theta}}, & \text{if } V_S < a \\ (b - V_S) \frac{\partial S_p}{\partial \boldsymbol{\theta}}, & \text{if } V_S > b \\ 0, & \text{otherwise} \end{cases} \tag{4}$$

where $\frac{\partial S_p}{\partial \boldsymbol{\theta}}$ denotes the standard derivative of the softmax outputs of the network. The gradient in (4) has a clear interpretation. During back-propagation, when the current constraints are satisfied, i.e., $a \leq V_S \leq b$, observe that $\frac{\partial \mathcal{C}(V_S)}{\partial \boldsymbol{\theta}} = 0$. Therefore, in this case, the gradient stemming from our term has no effect on the current update of the network parameters. Now, suppose without loss of generality that the current set of parameters $\boldsymbol{\theta}$ corresponds to $V_S < a$, which means the current target region is smaller than its lower bound $a$. In this case of constraint violation, term $(a - V_S)$ is positive and, therefore, the first line of (4) performs a gradient *ascent* step on softmax outputs, increasing $S_p$. This makes sense because it increases the size of the current region, $V_S$, so as to satisfy the constraint. The case $V_S > b$ has a similar interpretation.

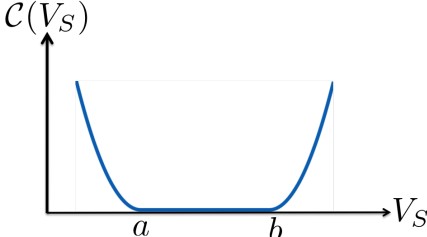

Figure 1: Illustration of our differentiable loss for imposing soft size constraints on the target region.

The next section details the dataset, the weak annotations and our implementation. Then, we report comprehensive evaluations of the effect of our constrained-CNN losses on segmentation performance. We also report comparisons to the Lagrangian-based constrained CNN method in [17] and to the fully supervised setting.

---

[3]The code can be found at https://github.com/LIVIAETS/SizeLoss_WSS

# 3 Experiments

## 3.1 Dataset

Our experiments focused on left ventricular endocardium segmentation. We used the training set from the publicly available data of the 2017 ACDC Challenge[4]. This set consists of 100 cine magnetic resonance (MR) exams covering well defined pathologies: dilated cardiomyopathy, hypertrophic cardiomyopathy, myocardial infarction with altered left ventricular ejection fraction and abnormal right ventricle. It also included normal subjects. The exams were acquired in breath-hold with a retrospective or prospective gating and a SSFP sequence in 2-chambers, 4-chambers and in short-axis orientations. A series of short-axis slices cover the LV from the base to the apex, with a thickness of 5 to 8 mm and an inter-slice gap of 5 mm. The spatial resolution goes from 0.83 to 1.75 $mm^2$/pixel.

For all the experiments, we employed the same 75 exams for training and the remaining 25 for validation. To increase the variability of the data, we augmented the dataset by randomly rotating, flipping, mirroring and scaling the images.

## 3.2 Weakly-annotated labels

To generate the weak labels, we employed binary erosion on the fully annotations with a kernel of size $10 \times 10$. If the resulted label disappeared, we repeated the operation with a smaller kernel (i.e., $7 \times 7$) until we get a small contour. Thus, the total number of annotated pixels represented the 0.1% of the labeled pixels in the fully supervised scenario. Figure 2 depicts some examples of fully annotated images and the corresponding weak labels.

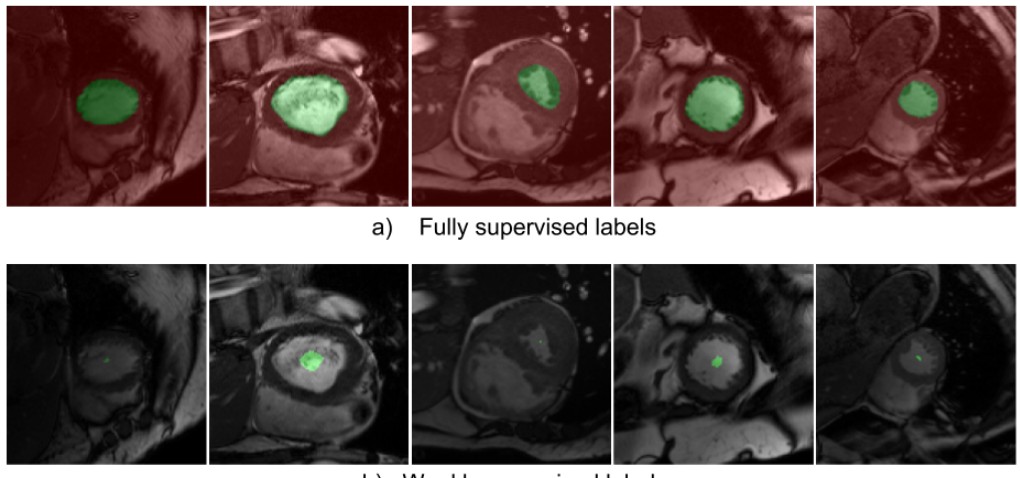

Figure 2: Examples of different levels of supervision. In the fully labeled images (*top*), all pixels are annotated, with red depicting the background and green the object of interest). In the weakly supervised cases (*bottom*), only the labels of the green pixels are known. The images were cropped for a better visualization of the weak labels. The original images are of size $256 \times 256$ pixels.

To compute lower and upper bounds on the size of the target region, we used the manual segmentation of a single subject. Specifically, we computed the minimum and maximum values of the size of left ventricular endocardium over all the slices, and multiplied by a factor of 0.9 and 1.1 the minimum and maximum value, respectively, to account for size variations across exams.

## 3.3 Several levels of supervision

Training models with diverse levels of supervision requires defining appropriate objectives for each case. In this section, we introduce the different models, each with different levels of supervision.

---

[4]https://www.creatis.insa-lyon.fr/Challenge/acdc/

First, we trained a segmentation network from weakly annotated images with no additional information, which served as baseline. Training this model relies on computing the cross-entropy only on labeled pixels, i.e., *partial cross-entropy (CE)*. Then, we trained several models using the same weakly annotated images with different levels of constraint supervision. First, similar to MIL scenarios, we used image-tag priors by enforcing the presence or absence of a the target in a given training image, as introduced earlier. This reduces to enforcing that the size of the predicted region is less or equal to 0 if the target is absent from the image, or larger than 0 otherwise. Afterwards, we imposed an upper-bound constraint on the size of the target when the latter is present in the image. In this case, $a$ in Eqs. (3) and (4) is set equal to 0. Thus, the CNN is constrained to generate segmentations whose sizes must be below an upper bound $b$. If the target is not present on the image, we set the upper bound $b$ equal to 0, similarly to the MIL case for suppression constraints.

Finally, we imposed both upper- and lower-bound constraints on the size of the target region. Forward and backward passes are defined by Eqs. (3) and (4), whith $a$ and $b$ computed from a single fully annotated subject, a described in Section 3.2. The lower and upper bounds were set equal to 98 and 1723 pixels, respectively. In all these cases, the partial cross-entropy of annotated pixels and the proposed constrained-CNN losses are jointly minimized. Finally, we report the full-supervision performance, where class labels (i.e., endocardium and background) are known for every pixel during training, and were used in a sum of per-pixel cross-entropy.

The Lagrangian proposals in [17] reuse the same network and loss function as the fully-supervised setting. At each iteration, the method alternates two steps. First, it synthesizes a ground truth $\tilde{Y}$ with projected gradient ascent (PGA) over the dual variables, with the network parameters fixed. Then, for fixed $\tilde{Y}$, the cross-entropy between $\tilde{Y}$ and $S$ is optimized as in standard CNN training. We found that limiting the number of iterations for the PGA to 500 (instead of the original 3000) saved time without impacting the results. We used the same bounds as in the proposed direct losses.

### 3.4 Training and implementation details

For all the experiments, we used ENet [15], as it has shown a good trade-off between accuracy and inference time. Nevertheless, the proposed loss is general and can be used in conjunction with any CNN. The network is trained from scratch by employing Adam optimizer and a batch size of 1. The initial learning rate was set to $5\times10^{-3}$ and decreased by 2 after 100 epochs. The weight of our loss in (2) was empirically set to $1\times10^{-2}$. The input images are of size $256 \times 256$ pixels. We computed the common Dice similarity coefficient (DSC) to compare the different models. We used a combination of PyTorch [16] and NumPy for our implementations, and ran the experiments on a machine equipped with a NVIDIA GTX 1080 Ti GPU (11GBs of memory). The code is available at https://github.com/LIVIAETS/SizeLoss_WSS.

## 4   Results

This section reports our results. First, in Sec. 4.1, we evaluate the impact of including different losses during training in a weakly supervised setting. Then, in Sec. 4.2, we juxtapose the performances of our direct loss to the iterative Lagrangian proposals [17], showing that our simple method yields substantial improvements over [17] in the same weakly supervised settings. We also provide the results for the fully supervised setting in Sec. 4.3. We further provide qualitative results in Sec. 4.4. Finally, we compare the different learning strategies in terms of efficiency (Sec. 4.5), showing that our direct constrained-CNN loss does not add to the training time, unlike the Lagrangian method in [**?** ].

### 4.1   Weakly supervised segmentation with size loss

Table 1 reports the results on the validation set for all the models trained with both the Lagrangian proposals in [17] and our direct loss. As expected, using the partial cross entropy with a fraction of the labeled endocardium pixels yielded poor results, with a mean DSC less than 0.10. Enforcing the image-tag constraints, as in the MIL scenarios, increased substantially the DSC to a value of 0.7122. More interestingly, we observed that constraining the CNN predictions with the proposed direct size loss significantly increased the performances, as can be seen on both Table 1 and Fig 3. Imposing both lower and upper bounds on the size of the predicted region yielded the best performance. Specifically,

if we use only an upper bound $b$ on region size, we achieve a mean DSC value of $0.8189$, whereas if a lower bound $a$ is also considered, mean DSC increases up to $0.8415$.

Table 1: Segmentation results with different levels of supervision.

| Model | Model | Method | DSC (Val) |
|---|---|---|---|
| | Partial CE | | 0.0721 |
| | CE + Tags | Lagrangian Proposals [17] | 0.6157 |
| | Partial CE + Tags | Direct loss (Ours) | 0.7122 |
| Weakly supervised | CE + Tags + Size* | Lagrangian Proposals [17] | 0.6175 |
| | Partial CE + Tags + Size* | Direct loss (Ours) | 0.8189 |
| | CE + Tags + Size** | Lagrangian Proposals [17] | 0.6526 |
| | Partial CE + Tags + Size** | Direct loss (Ours) | **0.8415** |
| Fully supervised | Cross-entropy | | 0.9284 |

*Upper bound / ** Lower and upper bounds

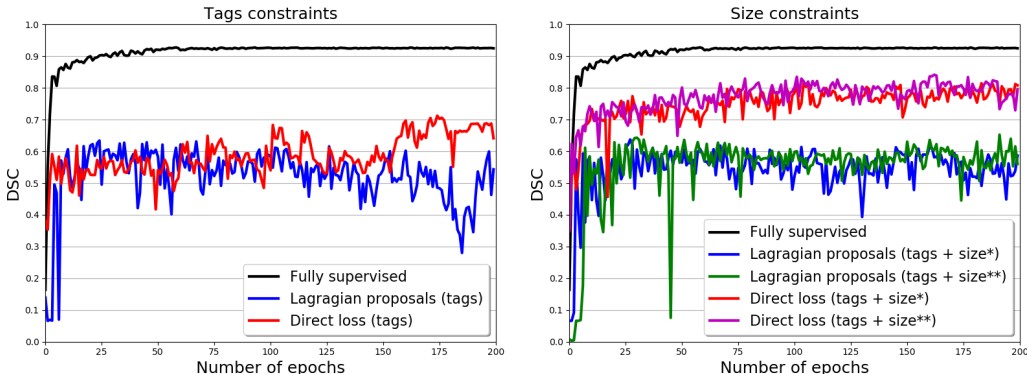

Figure 3: Evolution of the DSC on validation samples for the weakly supervised learning models and strategies analyzed, as well as for the full-supervision setting. We plot the evolution when using only image tags and when adding bounds on the size of the prediction.

## 4.2 Implicit loss against segmentation proposals

In this section, we juxtapose our direct constrained-CNN loss to the Lagrangian proposals in [17]. From Table 1 and Fig. 3 we observe that, for all the models of weak supervision, our direct loss outperformed substantially the Lagrangian proposals [17]. First, we observed that, for embedding image tags as suppression/presence constraints, Lagrangian proposals achieved a mean DSC of $0.6157$, while our direct loss yielded an increase of $15\%$, with a mean DSC equal to $0.7122$. This difference becomes more significant when adding constraints in the form of upper/lower bound on the size of target region. In these cases, our direct loss outperformed Lagrangian proposals by a margin of $30\%$. Another observation from Fig. 3 is that training with Lagrangian proposals typically achieves its best performance faster than the proposed loss term, but does not improve after a few epochs. More importantly, one can see that the DSC evolution for Lagrangian proposals is less stable than with our direct loss.

## 4.3 Pixel-level annotations

Finally, we compared the weakly supervised strategies to a network trained on fully labeled images. Training the network with strong pixel-level annotations achieved a mean DSC of $0.9284$ on the validation set. This result represents an increase of a $10\%$ with respect to the best proposed method. Nevertheless, it is worth mentioning that the performance achieved by our weakly supervised learning approach with size loss constraints approaches the fully-supervised setting with only 0.1 % of the annotated ground-truth pixels. This indicates that the current work has the potential to bridge the gap between fully and weakly supervised learning for semantic medical image segmentation.

## 4.4 Qualitative results

To get some intuition about the different learning strategies and their effects on the segmentation, we visualize some results in Fig. 4. Particularly, the model containing the partial cross-entropy and size loss constraints with lower and upper bounds is investigated. For the three learning methods, i.e., fully supervised, Lagrangian proposals and our direct loss, we selected the best performing model to display these images. Even though generating proposals during training actually improved the segmentation performance compared to the partial cross-entropy (Table 1), looking at the examples in Fig. 4 (*third column*), we can observe that these segmentations are far from being satisfactory. Nevertheless, integrating the proposed size-constrained loss directly in the back-propagation increases substantially the accuracy of the network, as can be seen in the last column of Fig. 4. An interesting observation is that, in some cases (*last row*), weakly supervised learning produces more reliable segmentations than training with full supervision.

| Ground truth | Fully supervised | Proposals | Ours |

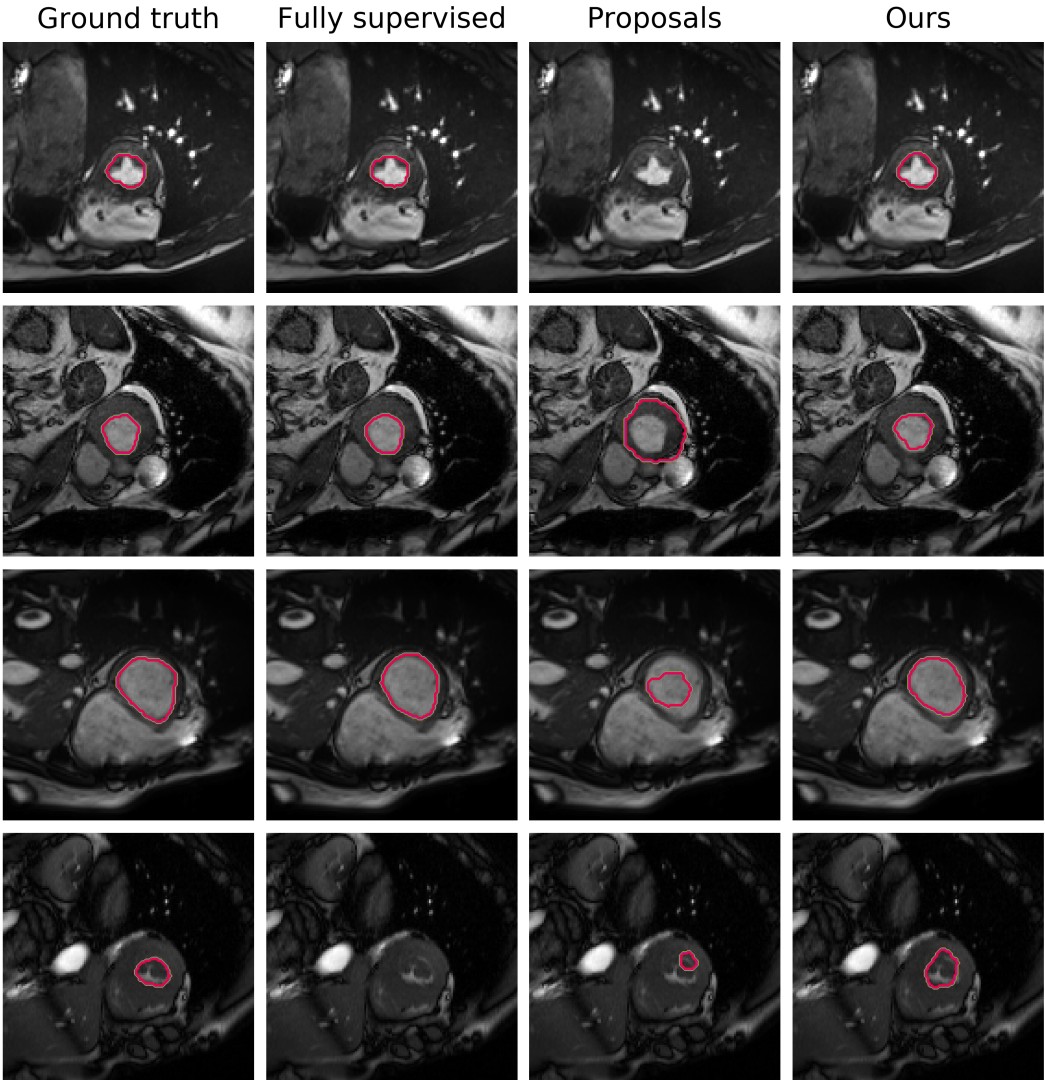

Figure 4: Qualitative comparison of the different methods. Each row represents a 2D slice from different scans. We can easily see that our method achieves results comparable to full supervision. (Best viewed in colors)

### 4.5 Efficiency

In this section, we compare the several learning approaches in terms of efficiency (Table 2). Both the weakly supervised partial cross-entropy and the fully supervised model need to compute only one loss per pass. This is reflected in the lowest training times reported in the table. Including the size loss does not add to the computational time, as can be seen in these results. As expected, the iterative process introduced by [17] at each forward pass adds a non-negligible overhead during training. To generate their synthetic ground truth, they need to optimize the Lagrangian function with respect to its dual variables (Lagrange multipliers of the constraints), which requires alternating between training a CNN and Lagrangian-dual optimization. Even in the simplest optimization case (with only one constraint), where optimization over the dual variable converges rapidly, the method still adds a few milliseconds at each iteration.

Table 2: Training times for the diverse supervised learning strategies, using tags and size constraints.

| Method | Training time (ms/image) |
|---|---|
| Partial CE | 10 |
| Direct loss | 10 |
| Lagragian proposals | 15 |
| Fully supervised | 10 |

## 5 Conclusion

We presented a novel loss function for weakly supervised image segmentation, which despite its simplicity reach almost full supervision performances. We perform significantly better than the other proposed methods for this task, achieving 90% of full supervision performance with only 0.1% of annotated pixels, while having negligible computation overhead. While our experiments focused on basic linear constraints such as the target-region size and image tags, our direct constrained-CNN loss can be easily extended to other non-linear constraints, e.g., invariant shape moments [5] or other region statistics [8]. Therefore, it has the potential to close the gap between weakly and fully supervised learning in semantic medical image segmentation.

**Acknowledgments**

This work is supported by the National Science and Engineering Research Council of Canada (NSERC), discovery grant program, and by the ETS Research Chair on Artificial Intelligence in Medical Imaging.

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
