# OpenReview forum: "Size-constraint loss for weakly supervised CNN segmentation"
_MIDL.amsterdam/2018/Conference — MIDL 2018 Oral_

### Review · AnonReviewer1 · 2018-04-21
**Review of Size-constraint loss for weakly supervised CNN segmentation**

**Rating:** 5
**Confidence:** 2

**Review:**

Why are there strong differences of accuracies between loose and tight bounds?
What's the reason of higher accuracy with two bounds?
What's standard deviation of DSC in each method?
You need to test with independent test set and extra-validation to evaluate overfitting of this model in this dataset only.

**Special Issue:**

Yes

---

### Review · AnonReviewer3 · 2018-05-09
**Nice method for weakly-supervised segmentation of biomedical images**

**Rating:** 4
**Confidence:** 2

**Review:**

Summary:
For weakly supervised segmentation on 3D images using CNN, a new loss function was proposed. The loss function has two parameters defining the range of zero-response, and become higher outside. The performance is shown using cardiac MR images.

Pros:
* New loss function specified for the problem.
* Good segmentation performance compared to the baseline.
* Public dataset is used and made their code publicly available

Cons:
* Way to choose values for a and b should be explained more.
Sec 4.1 says "a and b are computed from one training subject", but how?
* Discussion about the sensitivity to the choice of a and b of the results is missing.
* "The baseline is the network with cross entropy loss only on labeled pixels", page 5.
Currently, this sounds like unfair comparison with your method because it is theoretically apparent that the baseline does not work well, as the next line describes. Think about adding other comparison targets for good comparison. They don’t need to be based on deep learning.
* Paper structure is difficult to read (see below).

Overall opinion:
There are few works for semi-supervised segmentation on biomedical images. This paper shows that good performance can be obtained with CNN and their proposed loss function. This paper has good novelty for publication.

Specific comments:
* Paper structure: for better readability, describe what you have done in each evaluation topic must be explained in Section 3, and show results in Section 4. Another idea is to bind Sec 3 and 4 as "Materials and methods".
* Describe the baseline more in detail.
* Eq. (3) and text around it
  Assign a symbol for   \sum_{p \in \Sigma} S_p    for simplifying
* Last of 1.1
  - Footnote is empty. But URL is also written in last of 3.3
* In 3.3
  - e in "5e-3" and "1e-2" looks like the Napier's constant.
  - * (asterisk) "... < 6 * b" looks like the convolution.
  - Show versions of PyTorch and Numpy, and add name and version of operating system.
* Table 1:
  - Caption should be upper than the table body. Make the caption more detail.
* Figure 4:
  - What is the difference between rows?  Slices from common or different patients?
  - Add a column of weakly supervised labels.


**Special Issue:**

Yes

---

### Review · AnonReviewer2 · 2018-05-09
**Size-constraint loss for weakly supervised CNN segmentation**

**Rating:** 4
**Confidence:** 3

**Review:**

The paper presents a method based on convolutional network trained in a supervised fashion to address lesion segmentation in medical imaging when only limited amount of labels is available, namely in a weakly supervised segmentation setting.
The problem is tackled by proposing a novel loss term that takes into account for lower and upper bound of target lesions.
The method is validated on a publicly available dataset from the 2017 ACDC Challenge.
Results show that the proposed approach substantially improve the performance of a plain weakly supervised segmentation approach (solely using few annotated voxels) and gets closer to the performance of a fully supervised method (Table 1).
The paper is well written, the approach is clearly explained, and the validations is thorough.

The key component of the proposed loss function is the presence of a lower and upper bound.
Implicitly, this means that this loss function is somehow limited to segmentation problems where lesions with a specific size/shape have to be segmented.
It would be interesting to explore whether this approach can be further developed to tackle weakly supervised semantic segmentation, where the target is a dense prediction of multiple tissue types (and some of those could not have a well defined size/shape).

**Special Issue:**

Yes

---

### Comment · ~Bram_van_Ginneken1 · 2018-05-18
**Selection for longlist for special issue Medical Image Analysis**

Dear authors,

Congratulations on your acceptance to MIDL! We have selected your paper on the longlist for the Medical Image Analysis Special Issue. Please read this page:
https://midl.amsterdam/special-issue-in-medical-image-analysis/
Please answer the three questions that are listed on that page about your interest in submitting to the special issue, potential overlap with other publications, and related publications.

You can post your answer here directly below on openreview.net, or mail me directly at bram.vanginneken@radboudumc.nl.

Best regards, Bram

---

### Decision · Program_Chairs · 2018-05-15
**Paper112 Acceptance Decision**

Oral